# The Age-Related Performance Decline in Ironman 70.3

**DOI:** 10.3390/ijerph17062148

**Published:** 2020-03-24

**Authors:** Kristian Jäckel, Caio Victor Sousa, Elias Villiger, Pantelis T. Nikolaidis, Beat Knechtle

**Affiliations:** 1Medbase St. Gallen Am Vadianplatz, 9001 St. Gallen, Switzerland; Kristian.Jaeckel@medbase.ch; 2College of Arts, Media & Design, Bouve College of Health Sciences, Northeastern University, Boston, MA 02115, USA; cvsousa89@gmail.com; 3Institute of Primary Care, University of Zurich, 8091 Zurich, Switzerland; evilliger@gmail.com; 4Exercise Physiology Laboratory, 18450 Nikaia, Greece; pademil@hotmail.com

**Keywords:** half ironman, ironman 70.3, age-dependent performance decline, masters athletes, triathlon, endurance, multi-sports, split disciplines

## Abstract

Although the age-related decline in sport events has been well studied, little is known on such a decline in recreational triathletes for the Half Ironman distance. Indeed, the few existing studies concentrated on specific aspects such as top events, elite groups, some consecutive years, single locations, or age categories instead of analyzing all the data available. Therefore, the aim of the present study was to examine recreational triathletes’ performance in three split disciplines (swimming, cycling, and running) as well as in overall race time by analyzing all data of Half Ironman finishers found on ironman.com (i.e., 690 races; years 2004 through 2018; 206,524 women (24.6%) and 633,576 men (75.4%), in total 840,100 athletes). The age-dependent decline in Half Ironman started earliest in swimming (from the very first age group on) with a smallest age group delta between 35–49 years in men and 40–54 years in women. The performance decline started at 26 and 28 years in men and women for running; at 34 years for men and 35 years for women in cycling; and at 32 years for men and 31 years for women with regard to overall race time. The results may be used by coaches and recreational athletes alike to plan a triathlon career.

## 1. Introduction

Triathlon, a multidiscipline endurance sport (i.e., swimming, cycling, and running), has several distances: the Olympic distance (1.5 km swimming, 40 km cycling, and 10 km running), the Half Ironman distance (1.9 km/1.2 miles swimming, 90 km/56 miles cycling, and 21.1 km/13.1 miles running, the full Ironman distance (3.8 km swimming, 180 km cycling, and 42.2. km running), and distances longer than the Ironman distance, so-called ultra-triathlon events [1,2]. Triathlon performance is influenced by different factors such as personal best time, previous experience, sex, training, nationality, anthropometrical and physiological characteristics, pacing, performance in split disciplines, and age, which is considered the most important predictor of performance [3,4,5,6].

Peak performance in many sports—except the most explosive ones, which peak much earlier—is observed just before 30 years [7]. In Ironman triathlon, though, performance seems to plateau until approximately 35 years to be followed by a moderate negative slope for the next two decades and afterward thought to be steep [4,8,9,10,11]. This decline seems to be more prominent in women than in men, in longer distances than in shorter ones [9] as well as in off-road races than in road-based events [8,11]. It differs between split disciplines, which is due to their specific demands and because they differently challenge an aging athlete’s physiology, whereby the swim decline seems to be highest (when the focus is laid on recreational athletes) [12], followed by running decline, with cycling being easiest to keep performance in, as a study on the top 20% of non-elite athletes shows [11].

The age, at which age groups in each split discipline start to deteriorate, depends on the cohort investigated. For example, Lepers et al. [10,13] investigated swimming, cycling, running, and overall race times of the top ten male triathletes between 20 and 70 years of age (in 5-year intervals) for two consecutive Ironman World Championships (2006 and 2007). Their results showed a smaller age-related decline in cycling performance compared with that in running and swimming after 50 years of age for the Ironman distance. With advancing age, the performance decline was less pronounced in cycling (>55 years) and running (>50 years) [14]. On the other hand, according to another study on split and overall race times of 329,066 male and 81,815 female athletes from over 253 different Ironman triathlon races, the female age-related swim decline started much earlier, in the age group 25–29 years, and in the age group 30–34 years for cycling, running, and overall race time, whereas for men it started at the age group 25–29 years in swimming as well as in the age group 35–39 years in cycling, running, and overall race times [12]. Obviously, the selection of top athletes in elite races led to a considerable bias, which is revealed when considering all finishers of all races held worldwide.

The age of the fastest finishers in the “Ironman Hawaii” (i.e., the Ironman World Championship) has been increasing within the last decades. Meanwhile, these athletes’ overall race time improved [4,9,15]. Similarly, non-elite elderly age group (so-called masters) athletes have improved their performance in the “Ironman Hawaii” [9,15]. Masters athletes are typically defined as being older than 35–40 years [15,16] or as athletes of any age older than when world records are usually won [4,17]. There is a large increase in masters athlete participation rates over the last three to four decades [18].

Masters athletes gain increasing scientific interest. Their ability to maintain performance might help us understand health promotion better. Although a lot of research has been performed with respect to how age influences athletic performance in general and endurance events in particular, little data exist on its impact on performance in the Half Ironman triathlon, a relatively young sports event. Thus far, three studies have investigated age-related Half Ironman triathlon performance declines [9,18,19], focusing on aspects such as top events (i.e., World Championships or Olympic events [18]), elite groups (i.e., the top ten [9]), some consecutive years [18], or single locations [11,18,19]. Another one uses 10-year instead of 5-year age group categories [11]. This gap in turn means that we do not know how the normal recreational athlete might perform. Non-elite athletes are highly heterogenous and differ in many aspects—training status [20], nutritional status [21], experience [22,23]. Although the performance density of elite athletes is higher and therefore more reliable than that of recreational athletes, to our understanding we have no better data to extrapolate future recreational performance. This is due to the fact that elite and non-elite athletes cannot be compared due to several differences among the groups, most of all that elite athletes are younger than masters athletes, but that there are also differences in nutritional and training status as well as in mental preparation and strength (see also Section 4.10—limitations, practical applications, and implications for future research).

To the best of our knowledge no study exists so far that investigated the data of all Half Ironman races available on the official website ironman.com. In mass endurance events, only about 1% of the athletes are elitist. This number is taken from an Ironman study [12] but might serve as a rule for Half Ironman as well. As almost every athlete of our dataset taken from ironman.com is non-elitist (i.e., recreational), and many herein are amongst the group of masters athletes, we decided to study this important group. This approach might offer coaches and athletes the ability to plan a career and extrapolate the time as to when to change to a longer distance since it is known that older athletes place better in longer races [9].

We expected the age-related performance decline in Half Ironman races to start earlier than in Ironman triathletes according to Knechtle et al. who investigated the Olympic, Half Ironman, and Ironman distance; Wu et al., who report results for the same distances; and Käch et al. studying the decline pattern in triathlon [9,11,12]. Furthermore, it was assumed that recreational Half Ironman performance would follow these patterns: (i) swimming decline starting first, followed by running with cycling coming last (referring to the study by Käch et al. on recreational triathletes [12]); and (ii) the respective decline curves in women starting earlier and being more prominent than those of men [12]. This study might serve to increase our understanding of the performance decline in recreational Half Ironman triathlon that comes with aging.

## 2. Materials and Methods

### 2.1. Ethical Approval

This study was approved by the Institutional Review Board of Kanton St. Gallen, Switzerland, with a waiver of the requirement for informed consent of the participants as the study involved the analysis of publicly available data.

### 2.2. Data Sampling

We analyzed successful finishers of all Ironman 70.3 races recorded on the Ironman’s website (www.ironman.com/im703-races) between 2004—the first year under the official Ironman label—and 2018. The complete dataset of races with the official Ironman 70.3 label has, to the best of our knowledge, not yet been investigated. A total of 690 races (327 in North America, 67 in South America, 144 in Europe, 71 in Asia, 53 in Oceania, 7 in the Middle East, and 6 in Africa) were identified by using a self-programmed Visual Basic script facilitating data sampling.

Some races’ report data had to be transformed from PDF to Excel format using pdftoexcel.com. Some of these Excel-converts had to be adjusted to the correct order by hand. Transition times have not been considered. Exclusion criteria were (i) athletes who did not start; (ii) disqualified athletes; (iii) athletes with at least one missing split time; (iv) athletes who did not finish; (v) inappropriate time for an age group (i.e., split time much faster than the appropriate time of the winner of the race, possibly due to technical recording issues); (vi) elite athletes; (vii) athletes aged 75 years old or older. In the end, 206,524 women (24.6%) and 633,576 men (75.4%) were included, totalizing 840,100 athletes’ data as the basis for the following results.

### 2.3. Statistical Analysis

The data were tested for normality and homogeneity with Kolmogorov–Smirnov and Levene’s tests, respectively. Two-way ANOVA was applied to compare performance data of swimming, cycling, running, and overall using sex as a fixed factor and age group as a random factor (age groups M20 and 75 and above were excluded because of insufficient sample). Delta percentage (Δ%) was calculated between adjacent age groups for men and women. Non-linear regression analysis was applied in performance × age in each split and overall race time. Scheffe’s post hoc was applied. Statistical significance was set at 1% (α < 0.01). All procedures were made using Statistical Software for the Social Sciences (IBM, SPSS v20.0. Chicago, IL, USA) and GraphPad Prism (GraphPad Prism v6.0. San Francisco, CA, USA).

## 3. Results

An age effect was identified for both men and women for all three disciplines (*p* < 0.0001). The post hoc analysis showed all age groups to be statistically different from their previous one in all three disciplines and for both men and women (Figure 1a–c), i.e., the older the age, the slower the race time. Figure 1 presents performance decline with regard to age and shows age groups from 18–24 until 70–74 years. The three single-split discipline results are plotted in Figure 1a (swimming), Figure 1b (cycling), and Figure 1c (running). Figure 2 adds the respective delta between two adjacent age groups, and Figure 3 shows overall race time age group performance deterioration as well as overall delta change with age. Figure 1a presents swimming speed (km/h), which continuously decreases with age. Its smallest deterioration is between age groups 40–44 and 45–49 years for men (delta 0.76%) and between 45–49 and 50–54 years for women (delta 0.76%). Swimming performance analysis shows an effect for age group (F = 27.0; *p* < 0.001), sex (F = 190.7; *p* < 0.001), and interaction (F = 49.3; *p* < 0.001). Figure 1b shows cycling speed (km/h) for all age groups, which starts to deteriorate at 34 years in men and 35 years in women. Cycling performance analysis shows an effect for age group (F = 38.5; *p* < 0.001), sex (F = 2172.8; *p* < 0.001), and interaction (F = 22.8; *p* < 0.001). Figure 1c is about running speed (km/h). In men, performance decline starts at about 26 years, in women at 28 years. Running performance analysis shows an effect for age group (F = 97.7; *p* < 0.001), sex (F = 853.4; *p* < 0.001), and interaction (F = 20.8; *p* < 0.001).

Figure 2 describes how any two adjacent age groups differ in performance (i.e., performance delta). It is interesting to note here that the swimming delta (in Figure 2a) has its absolute minimum between 40–49 and 45–54 years in men and women, respectively. The constant positive delta (%) is higher in the older and the younger age groups (i.e., 55–74- and 18–39-year-old athletes being the highest between 65–69 and 70–74 years at 5.46% in men and 4.86% in women). Figure 2b on the cycling speed delta shows that the decline of all age groups increases except between 55 and 69 years in both sexes (especially in women). The running speed delta is (more or less) constantly deteriorating between age groups, as Figure 2c presents. Between the age groups 45–49 and 50–54 years, the delta rises significantly. From around 1.4% between 30 and 49 years, it increases to about 2.25% between 45 and 54 years and up to 6.85% between 65 and 74 years in men and from 1.74% over 2.52% until 6.01% in women. Both sexes deteriorate in a similar pattern.

Figure 3a on overall performance decline shows a constant deterioration in men and, from 25 to 34 years onward, in women as well (i.e., in women the age group 25–29 years is faster than the age group 18–24 (delta is −0.8%, see Figure 3b)). Overall performance analysis shows an effect for age group (F = 288.8; *p* < 0.001), sex (F = 3206; *p* < 0.001), and interaction (F = 6.7; *p* < 0.001). The age of peak performance for overall race time and each discipline by sex is presented in Table 1.

## 4. Discussion

This study attempted to identify how age groups of recreational master athletes perform in the Half Ironman triathlon. Hence, all athletes worldwide of all officially documented races available were considered. The main results were that (i) the swimming decline started from the youngest age group on, and that (ii) there was a sex difference in swimming and cycling, but not in running. Performance decline in multidiscipline sports depends on locomotion modes [9,11,12,13,24]. In accordance with Käch et al. [12], who investigated recreational Ironman triathletes, we expected that Half Ironman performance would follow these patterns: swimming decline starting first, followed by running and lastly cycling. Performance decline (at least in professionals) varies with race length and, thus, we expected it to be less prominent in Half Ironman than in Ironman triathletes [9,11].

### 4.1. Earlier Age-Related Decline in Swim Performance Compared to that in Cycling and Running

The first result was that the age-dependent decline in Half Ironman started earliest in swimming, where every age group was slower than its previous one. The smallest changes take place between the age groups 35–49 and 40–54 in men and women, respectively. Käch et al. also described such a constant decline for recreational Ironman triathletes for women but not for men, where the 25–29 age group was faster than the 18–24 age group (see Figure 1 in [12]). Besides this, there is almost always an inflection point in endurance sports performance decline (see Stones and Hartin [18]), as not only strength but also knowledge and experience matter (especially in ultra-endurance events), enabling older athletes to outperform younger ones. By contrast, in “elitist” groups, i.e., when investigating top 5 or top 10 athletes per groups, the swim decline seems to start at about 40–45 years of age, almost irrespective of race distance, namely in Olympic World Championships [10]; in Half Ironman (20 best athletes per age group in a qualifier race) [9,19]; and in "Ironman Hawaii” [10]. Swim age group decline, however, when studying larger groups, also started earlier than in the abovementioned elitist groups, namely from 30 to 34 years onward in 5549 athletes competing in Half Ironman World Championship races (between 2006 and 2010) [18] and, with an even stronger effect, when analyzing all races worldwide (i.e., constant deterioration from 18 to 24 years onward) in masters Ironmen [12]. The differences between elite and recreational athletes are obvious as their training differs concerning volume, intensity, and motivation.

### 4.2. Decline in Running Performance with Initial Plateau

In this study on the Half Ironman triathlon, running speed started to decrease at 26 years (men) and 28 years (women) (see Figure 1c and Table 1). Stones and Hartin [18] mention the Half Ironman age group decline to be progressive after the age of 30–34 years. Those differences might be because of the different numbers of athletes and race types investigated (i.e., World Championships and 5549 athletes by Stones and Hartin [18] compared to all races worldwide and 840,100 athletes in this study). A similar relation exists between the study by Lepers et al. [10] and Käch et al. [12]. Lepers stated a progressive running decline for World Champions after the age of 50 years in a study with a rather smaller number of athletes. By contrast, Käch et al. [12], who investigated recreational triathletes, reported a much earlier decline starting at about 30–34 years in women and 35–39 years in men. Elite athletes maintain performance up to the fourth or fifth decade of life, i.e., a curvilinear decline from 50 years onward in Olympic World Championships [10]; from 45 years onward for top athletes in a Half Ironman qualifier [9]; and from 45 years onward in the "Ironman Hawaii” [10]. Elite athletes are capable of maintaining their high performance over a longer period than recreational athletes because of their training being more intensive and high-volume oriented.

### 4.3. Cycling Performance Decline Is Least Pronounced Compared to Other Split Disciplines

Cycling age-dependent performance decline (at least at World Championship race level) is known to be less pronounced in shorter than in longer distances [10] with air resistance (and therefore cycle time as well) being the primary energy cost factor at high cycling speeds since mechanical power herein depends on the third power of speed [4,13]. Performance of recreational Ironman triathletes, who cycle 180 km per race, is known to start to deteriorate from 30 to 34 years onward in women and 35–39 in men [12]. In this study on races with a 90 km cycling split, the athletes’ performance started to decline at 23 years for women and at 25 years for men (Table 1). The discrepancy with the results of Lepers et al. could stem from them investigating professionals, in whom the abovementioned physiologic regularities are effective. However, in recreational age group master athletes, the performance density of all athletes of a given age group could be too wide to see such regularities work.

### 4.4. Overall Decline Starts Earlier than in Ironman

According to Käch et al., the performance of recreational Ironman athletes started to deteriorate at 30–34 years in women and at 35–39 years in men [12]. In this study on non-elite master athletes, the significant decline started at the age of 31 and 32 years for women and men, respectively (see Figure 3; Table 1). The Ironman decline taking place later than that in the Half Ironman is what literature suggests [9,11].

Stones and Hartin [18] reported that for the oldest age groups in the Half Ironman, cycling and running contributed for the most part to overall performance; approximate relative times spent in each subdiscipline were 12% (swimming), 51% (cycling), and 37% (running). Comparing Käch’s results [12] and ours, we have to state that the overall decline in Half Ironman triathlon starts earlier than that in Ironman.

### 4.5. Swim Performance Decline Does Significantly Differ between the Sexes

Surprisingly, sex difference was highest in swim performance decline. In endurance events, it is usually thought to be 10%–15%, mostly irrespective of exercise duration, widening with age [11], and it is usually thought to be smallest in swimming. Women have 7%–12% higher body fat than men [24] and there is less leg drag due to smaller body size, these factors, with other aspects, give women an advantage compared to men [19,24]. Others found the largest sex differences in short-distance triathlon swimming [11], where the abovementioned benefits should not pay off as well as in longer distances. Lepers stated a 12% sex difference in recreational triathletes [24]. Tanaka and Seals found a greater swim decline in women than in men [22], which goes along with our dataset, and Lepers and Maffiuletti [25] recorded that the sex difference in swimming narrowed with increasing race length—from about 19% in 50 m to approximately 11% in 1500 m events).

### 4.6. Cycling Performance Decline Does Significantly Differ between the Sexes

Age groups of recreational full-distance ironmen have a cycling sex difference of about 15% [24]. Most interestingly, men’s performance decline in this study is more pronounced than women’s (see Figure 2b), which we did not expect [12]. A potential explanation could be that a decline which starts at a higher level might deteriorate more pronouncedly compared to the one that starts on lower numbers [26].

### 4.7. Overall Performance Does Not Exhibit Significant Sex Differences

The age-related decline in overall race time seemed to be greater in women than in men, in the elderly than in the young [19], and usually less pronounced in shorter than in longer distances [15]. In Ironman, the overall sex difference narrowed from 15% (1990s) to 11% (2012) with women’s running improvements being most responsible for that [24]. These findings line up with our results (see Figure 3b).

### 4.8. Potential Physiological Explanations for the Age-Related Decline in Performance

Many factors affect age-dependent declines. Individual medical condition (e.g., disease, trauma sequelae) as well as physiological factors (e.g., maximal oxygen consumption (VO_2_max), lactate threshold and exercise economy [16]), which deteriorate while aging, may slow down individuals and therefore have group effects. VO_2_max seems to be the most important physiological factor for women as well as men [12]. Peak and mean power output depend on lower-limb muscle mass [19], which in turn explains why women usually have greater decline rates than men since cycling and running account for about 90% of race time [18]. Women’s best times may be predicted by many aspects, i.e., relative maximum oxygen uptake while swimming, overall speed in running at ventilatory threshold, and a few more, whereas men’s only significant predictor of overall race time was running speed at ventilatory threshold [19].

Another perspective suggested by Lepers [15] differentiates between central physiology (e.g., maximum heart rate, maximum stroke volume, blood volume) and peripheral physiology (e.g., muscle mass, muscle fiber composition, etc.), which might be trained [15], with older women’s exercise volume/intensity maybe being smaller than older men’s [9]. Sociodemographic factors probably also account for the decline seen in this study [15]. The Half Ironman triathlon, being a young and emerging sport might have more less-trained masters athletes than Ironman events, as participation in an Ironman requires better preparation [4]. Decline deceleration in endurance events mostly depends on experience [12,18], learning, applying, and fine tuning race tactics over the years [4], as well as being aware of and able to deal with psychological factors that enhance or dampen motivation [12]. Better training knowledge (which is easily accessible via the internet) and opportunities (which are easy to partake in for both sexes in a sports-centered century) enable all recreational athletes to perform better than in previous years.

### 4.9. Characteristics, Strengths, and Weaknesses (Biases) of this Study

We tried to avoid an “elite bias” because only about 1% of participants are elitist in mass endurance events (a number taken from an Ironman study) [12]. The study therefore comprises a large dataset of 840,100 athletes (24.6% female and 75.4% male athletes) with its relative female participation being located within the lower normal range for endurance events [24]. The study’s design is cross-sectional, thus not allowing for longitudinal changes but considering age group changes. External conditions such as weather and racecourse characteristics have not been considered but should have exerted little influence due to the number of race events (which was 690). The graphs exhibit relatively high standard deviations, probably due to the high number of participants in Half Ironman events, widening performance variation.

### 4.10. Limitations, Practical Applications, and Implications for Future Research

Considering the large number of triathletes participating in various race distances of triathlon—including the Ironman 70.3—and the variation of the age of these athletes, the results of the present study would be of great practical value for practitioners (e.g., coaches and fitness trainers) working in this sport. The large variation of ages implies that practitioners might work with groups of athletes demanding the formulation of individualized age-related training goals. Such age-related training goals would not only concern the overall race time, but also the performance in each discipline. In addition, the majority of triathletes might be considered as non-elite athletes with characteristics differing from their elite counterparts; thus, trends in participation and performance of elite athletes would not be valid for non-elite athletes highlighting the importance of the practical relevance of our findings. In this context, practitioners might use the data of this study as a reference to evaluate the overall race time and performance by discipline and set long-term training goals considering the expected age-related decline in performance.

External conditions (e.g., weather, training background) and motivational aspects have not been included in this analysis. As the Half Ironman has only been under the official label since 2004, it is a young phenomenon. It would therefore be interesting to understand the athletes’ motivation better, especially, what makes them choose this distance and train hard while getting older. A deeper understanding of the training habits of recreational Half Ironman triathletes is needed in order to compare their decline patterns better (comparing age group athletes with same training volume and intensity). A continuation of the study with appropriate physiological indicators (of a subgroup of its over 840,100 athletes) may be of benefit and add to our understanding of performance decline.

With regard to the magnitude of the relationship of race and split times with age, R^2^ was close to 1 denoting an almost perfect relationship. This observation might be partially attributed to the application of a second-degree polynomial regression analysis that seemed fitted with age-related decline in performance, i.e., relatively small differences among young age groups, whereas larger differences were shown among old age groups. This notion is in agreement with the theoretical knowledge suggesting a deeper decline in physiological characteristics related to performance after a certain age [4,8,9,10,11]. In addition, the R^2^ value might also be so high due to the use of a very large sample size that “masked” the influence of any outliers.

## 5. Conclusions

Performance deterioration in Half Ironman triathlete age groups as shown in this study begins at the first age group with swimming, at 28 (women) and 26 (men) years for running, at 34 (men) and 35 (women) years in cycling (as expected based on Käch et al. [12] for Ironman triathlon), and at 31 (women) and 32 (men) years of age concerning overall race time. Among these numbers, cycling does not seem to play as big a role as one might think because of its short relative duration during a Half Ironman race. Half Ironman athletes spend about 12% of time in swimming, 51% in cycling, and 37% in running [18]. In order to achieve best results, we conclude that every athlete should invest in swim training as early as possible to minimize any possible weaknesses that may inhibit best times. Cycling seems to be least prone to deterioration but still needs lots of training as it constitutes the biggest part of overall time (in order to reduce fatigue during and preserve a high pace for the run split). The strongest training emphasis though should be deployed to running, the second longest discipline, in which the biggest amount of fatigue has to be dealt with. In Ironman, cycling, running, and overall race time started to deteriorate at 30–34 years in women and at 35–39 years in men. The performance in swimming started to deteriorate at 25–29 years in both sexes [12]. Hence, it seems logical that women above 31 and men above 32 years should consider changing to Ironman in order to be able to achieve or preserve best results. In summary, this study’s results go along with literature showing that swim performance in recreational Half Ironman athletes seems to be hardest to preserve; that run performance decline is a constant one; and that cycling performance can be preserved up to older age groups.

## Figures and Tables

**Figure 1 ijerph-17-02148-f001:**
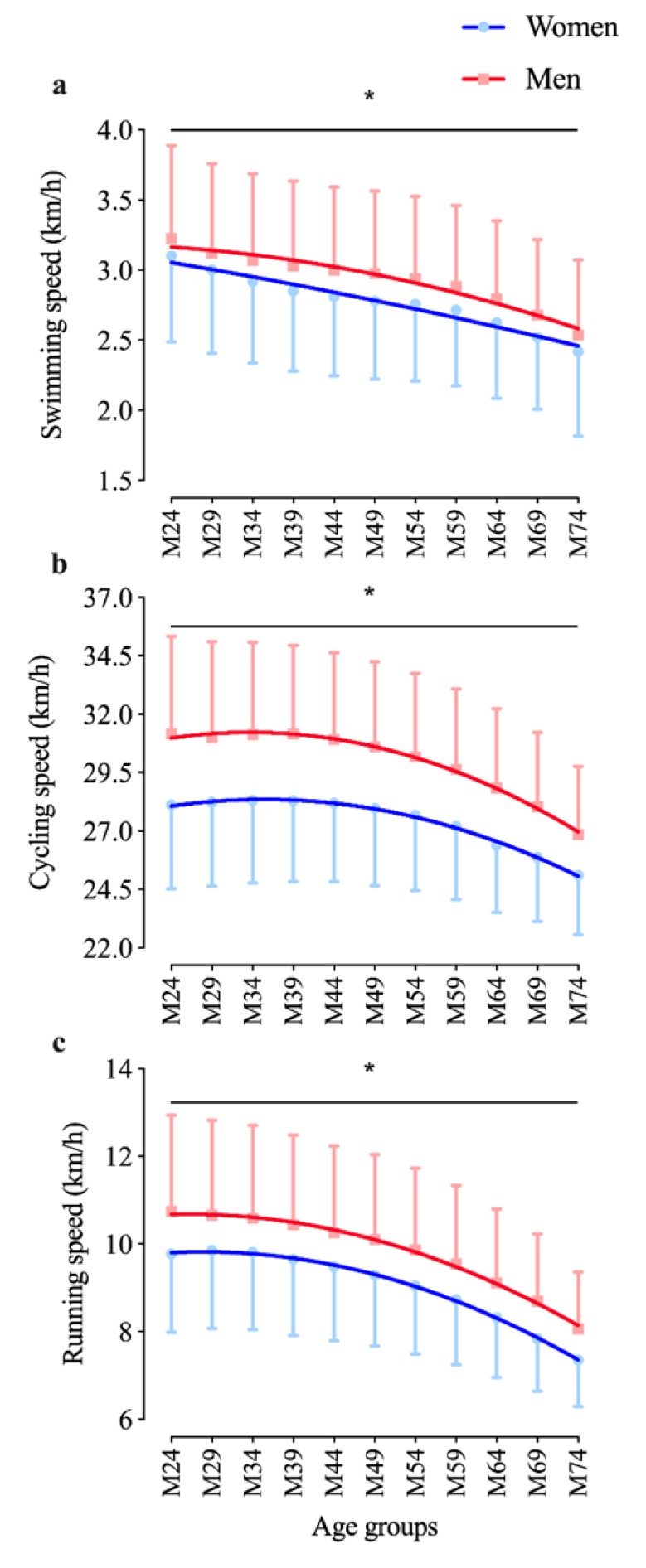
Age group-dependent decline in split discipline performance expressed in km/h; (**a**, top): swimming speed; (**b**, middle): cycling speed; and (**c**, bottom): running speed; (**a**–**c**): men in red, women in blue. * Age group, sex, and interaction effect (*p* < 0.05).

**Figure 2 ijerph-17-02148-f002:**
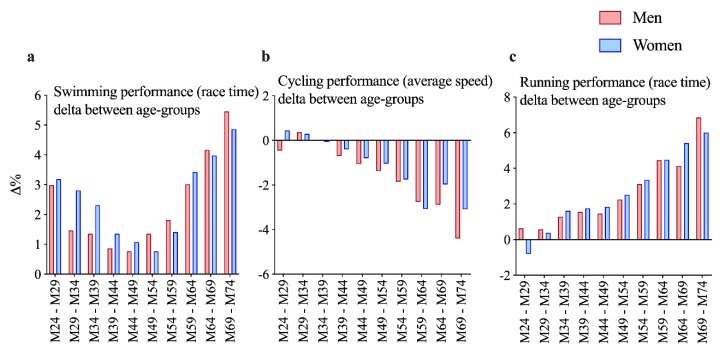
Split performance delta (%) between any two adjacent age groups; (**a**, on the left): swimming speed (km/h); (**b**, in the middle): cycling average speed (km/h); and (**c**, on the right): running speed (km/h); (**a**–**c**): men in red, women in blue.

**Figure 3 ijerph-17-02148-f003:**
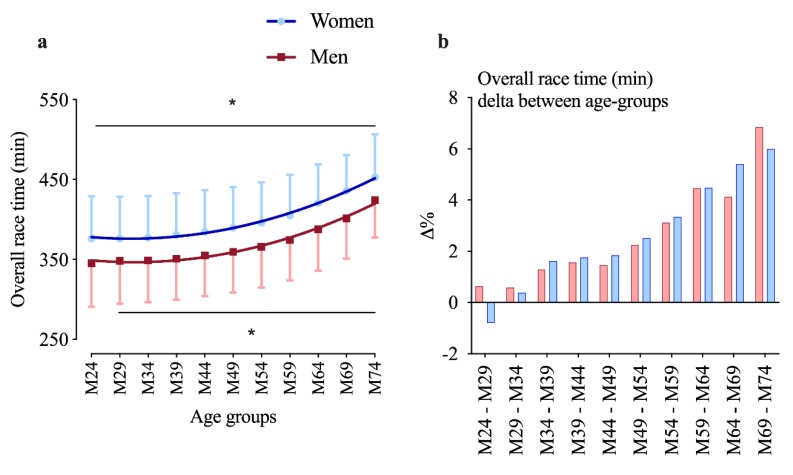
Overall race time; (**a**, on the left): overall race time for both sexes; (**b**, on the right): race time delta (%) between any two adjacent age groups; (**a**,**b**): men red, women blue. * Different from previous age group (*p* < 0.05).

**Table 1 ijerph-17-02148-t001:** Parameters in second-order polynomial regression Y = a + bx + cx^2^ using the race time of all athletes in 1-year age intervals by sex.

Sex and Discipline		Parameter			R^2^	Age (Years)	Race Time (min) or Speed (km/h)
		a	b	c			
Women							
	Swimming	3.265	−0.00776	−0.00004	0.968	-	-
	Cycling	25.59	0.1561	−0.00220	0.997	35.48	22.82
	Running	8.909	0.06502	−0.00116	0.999	28.03	8.00
	Overall	414.8	−2.513	0.04067	0.996	30.90	375.98
Men							
	Swimming	3.169	−0.00348	−0.00015	0.964	-	-
	Cycling	28.25	0.1765	−0.00262	0.996	33.68	25.28
	Running	9.941	0.05702	−0.00109	0.996	26.16	9.20
	Overall	386.9	−2.561	0.04063	0.990	31.52	346.54

Y = speed (km/h) in swimming, cycling, and running; Y = race time (min) in overall performance. For both approaches (i.e., 1 and 5 year) a second-degree polynomial analysis does not provide valid data as the youngest age group is the fastest one.

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
