# Peer review of "The Age-Related Performance Decline in Ironman 70.3"

_ijerph, 2020, doi:10.3390/ijerph17062148_

Round 1

Reviewer 1 Report

I suggest to increase the analyzes, perhaps in a continuation of the study, with physiological indicators that justify the decline in the performance of the amateur athletes, as well as to cross recovery data and life outside the sport that may be contributing to this decline.

Author Response

Reviewer 1

I suggest to increase the analyzes, perhaps in a continuation of the study, with physiological indicators that justify the decline in the performance of the amateur athletes, as well as to cross recovery data and life outside the sport that may be contributing to this decline.

Answer: We agree with the expert reviewer and changed the document text as following: A continuation of the study with appropriate physiological indicators may be of benefit and add to our understanding performance decline as well as would crossed data of recovery and leisure time do.

Reviewer 2 Report

This study determined the age-related decline in Half-Ironman performance from a very large database-- nearly 700 races and almost 900k athletes. The authors should be commended on this large under-taking. The manuscript is well-written and focused with clear messaging. 

However, the novelty of this investigation is unclear. The authors define a "gap" in the literature, however, provide no compelling rationale to investigate this gap nor any clear hypotheses of this investigation. Additionally, the merit and rationale of analyzing non-professional athletes (with potentially much greater heterogeneity than professional athletes) should be more clearly stated.

Author Response

Reviewer 2

This study determined the age-related decline in Half-Ironman performance from a very large database-- nearly 700 races and almost 900k athletes. The authors should be commended on this large under-taking. The manuscript is well-written and focused with clear messaging. 

However, the novelty of this investigation is unclear. The authors define a "gap" in the literature, however, provide no compelling rationale to investigate this gap nor any clear hypotheses of this investigation. Additionally, the merit and rationale of analyzing non-professional athletes (with potentially much greater heterogeneity than professional athletes) should be more clearly stated.

Answer: We agree with the expert reviewer that these points should be stated clearly but think that we addressed that issue: the lines 159-167 state the expectation of our studying the decline; the lines 58-76 summarize the most important aspect of literature, i.e. investigating elite vs. recreational athletes leading to different results; the line 210 states that the performance diversity in the recreational athletes’ group might be a possible bias factor and the lines 279/280 point towards the same direction.

Reviewer 3 Report

Congratulations for your job.

From my point of view and considering my possible limitations, I consider it a very well done job. Methodologically everything is correct and is expressed properly. It is easy to understand and attractive to the reader.

Author Response

Reviewer 3

Congratulations for your job.

From my point of view and considering my possible limitations, I consider it a very well-done job. Methodologically everything is correct and is expressed properly. It is easy to understand and attractive to the reader.

Answer: We thank the expert reviewer for his/her comment. No changes made.

Reviewer 4 Report

Accept.

Author Response

Reviewer 4

Accept

Answer: We thank the expert reviewer for his/her comment. No changes made.

Round 2

Reviewer 2 Report

This manuscript is one of several from this group investigating triathlon performance, and the incremental advance in knowledge from this investigation is unclear. 

Additionally, non-elite athletes are highly heterogenous differing in many aspects-- training status, nutritional status, experience, etc. (as stated by the authors)-- thus, these data are descriptive.

Thus, it appears this descriptive study is a small (if any) advance in knowledge relative to recent work from the authors (e.g. Kach et al 2018; Knechtle 2016; Etter et al 2013; Rust et al 2013; Knechtle et al 2012, etc.) and others (Lepers 2019; Lepers et al. 2013).

Although the novelty of the investigation is stated, there is no compelling rationale to investigate this novelty nor any potential impact of these data. The assertion that these data may be useful for coaches/athletes is fundamentally flawed, because of the heterogeneity in the athletes; data from elite athletes may be much more useful for coaches/athletes.

Author Response

Reviewer 2 replied:

“This manuscript is one of several from this group investigating triathlon performance, and the incremental advance in knowledge from this investigation is unclear. 

Additionally, non-elite athletes are highly heterogenous differing in many aspects-- training status, nutritional status, experience, etc. (as stated by the authors)-- thus, these data are descriptive.

Thus, it appears this descriptive study is a small (if any) advance in knowledge relative to recent work from the authors (e.g. Kach et al 2018; Knechtle 2016; Etter et al 2013; Rust et al 2013; Knechtle et al 2012, etc.) and others (Lepers 2019; Lepers et al. 2013).

Although the novelty of the investigation is stated, there is no compelling rationale to investigate this novelty nor any potential impact of these data. The assertion that these data may be useful for coaches/athletes is fundamentally flawed, because of the heterogeneity in the athletes; data from elite athletes may be much more useful for coaches/athletes.”

We agree with the reviewer that changes have to be made and changed the document as follows: Please see here the new lines 65 ff

Masters athletes gain increasing scientific interest. Their abilities to preserve performance might help us understand health promotion better. Although a lot of research has been performed with respect to how age influences athletic performance in general and in endurance events in particular, little data exists on its impact on performance in Half Ironman triathlon, a relatively young sports event. To this moment of time, three studies investigated age-related Half Ironamn triathlon performance declines [7,11,12], which have been focusing on aspects such as top events (i.e. World Championships or Olympic events [11]), elite groups (i.e. the top ten [7]), some consecutive years [11] or single locations [11,12,16]. Another one uses 10 year- instead of 5 year-age group categories [16]. This gap in turn means that we do not know how the normal recreational athlete might perform. Non-elite athletes are highly heterogenous and differ in many aspects – training status (J Sports Med Phys Fitness. 2011 Dec;51(4):583-94. The impact of physical, nutritional, and mental preparation on triathlon performance. Houston M, Dolan S, Martin S.), nutritional status (Many non-elite multisport endurance athletes do not meet sports nutrition recommendations for carbohydrates. Masson G, Lamarche B. Applied Physiology, Nutrition and Metabolism-7-41 (2016)), experience (J Sci Med Sport. 2014 May;17(3):300-5. doi: 10.1016/j.jsams.2013.04.014. Epub 2013 May 22. Predictive variables for half-Ironman triathlon performance. Gilinsky N, Hawkins KR, Tokar TN, Cooper JA), motivation (J Sci Med Sport. 2014 May;17(3):300-5. doi: 10.1016/j.jsams.2013.04.014. Epub 2013 May 22. Predictive variables for half-Ironman triathlon performance. Gilinsky N, Hawkins KR, Tokar TN, Cooper JA) (Sports (Basel). 2019 Sep 10;7(9). pii: E208. doi: 10.3390/sports7090208. Motivation Regulation among Black Women Triathletes. Brown CS)). Although the performance densitiy of elite athletes is higher and therefore more reliable than that of recreational athletes, to our understanding we have no better data to extrapolate future recreational performance as elite and non-elite athletes cannot be compared due to several differences among the groups most of all that elite-athletes are younger than Masters athletes but then also in nutritional and training status as well as mental preparatoin and strength (see also heading 4.10, limitations, practical applications and implications for future research). To the best of our knowledge no study yet exists which investigated the data of all Half Ironman races available on the official website ironman.com. In mass endurance events, only about 1% of the athletes is elitist. This number is taken from an Ironman study [17] but might serve as a rule for Half Ironman as well. As almost every athlete of our data set taken from ironman.com is non-elitist (i.e. recreational), and many herein are amongst the group of Masters athletes, we decided to study this important group. This approach might offer coaches and athletes the ability to plan a career and extrapolate the time as to when change to a longer distance as it is known that older athletes place better in longer races [7].

We expected the age-related performance decline in Half Ironman to start earlier than in Ironman triathletes according to Knechtle et al. who investigated the Olympic, Half Ironman and Ironman distance as well as to Wu et al., who report results for the same distances; and Käch et al. studying the decline pattern in triathlon [7,16,17]. Furthermore, it was assumed that recreational Half Ironman performance would follow these patterns: (i) swimming decline starting first, followed by running and cycling coming last (referring to the study by Käch et al. on recreational triathletes [17]); and (ii) the respective decline curves in women starting earlier and being more prominent than those of men [17]. This study might serve to increase our understanding the very performance decline in recrational Half Ironman triathlon that comes with aging.

Please have also a look the new lines 307 ff (we changed the heading to the following and made changes marked as red letters)

4.10. Limitations, practical applications and implications for future research

Considering the large number of triathletes participating in various race distances of triathlon – including the Ironman 70.3 version – and the variation of the age of these athletes, the results of the present study would be of great practical value for practitioners (e.g., coaches and fitness trainers) working in this sport. The large variation of ages implied that practitioners might work with groups of athletes demanding setting individualized age-related training goals. Such age-related training goals concerned not only the overall race time, but also the performance in each discipline. In addition, the majority of triathletes might be considered as non-elite athletes with different characteristics than elite athletes; thus, trends in participation and performance of elite athletes would not be valid for non-elite athletes highlighting the importance of practical relevance of our findings. In this context, practitioners might use the data of this study as reference to evaluate the overall race time and performance by discipline, and set long-term training goals considering the expected age-related decline in performance.

External conditions (e.g. weather, training background) and motivational aspects have not been included in this analysis. As Half Ironman is under the official label only since 2004, it’s a young phenomenon. It would therefore be interesting to understand the athletes’ motivation better, esp. what makes them choose this distance and train hard while getting older. A deeper understanding of the training habits of recreational Half Ironman triathletes is needed in order to compare their decline patterns better (compare age-group athletes with same training volume and intensity). A continuation of the study with appropriate physiological indicators (of a subgroup of its over 840,100 athletes) may be of benefit and add to our understanding performance decline as well as would crossed data of recovery and leisure time do.

There are other changes we’ve been marking throughout the whole document with red. Esp. one mistake had to be corrected: the overall decline starts earlier not later than in Ironman

4.4. Overall decline starts earlier than in Ironman

That made sequelae changes happen:

  1. Conclusions

Performance deterioration in age group Half Ironman triathletes as shown by this study begins at the first age-group with swimming, at 28 (women) and 26 (men) years for running, at 34 (men) and 35 years (women) in cycling (as expected based on Käch et al. [17]) and at 31 (women) and 32 (men) years of age concerning overall race time. Upon these numbers cycling doesn’t seem to play as big a role as one could think of because of its time spent on during a Half Ironman race. Age group Half Ironmen spend about 12 % with swimming, 51 % with cycling and 37 % with running [11]. In order to achieve best results, we conclude that every athlete should invest in swim training as early as possible to minimize any possible weaknesses that may inhibit best times. Cycling seems to be least prone for deterioration but still needs lots of training as it constitutes the biggest part of overall time (in order to reduce fatigue during and preserve a high pace for the run split). The strongest training emphasis though should be deployed to running, the second longest discipline, in which the biggest amount of fatigue has to be dealt with. In Ironman, cycling, running and overall race time started to deteriorate at 30-34 years in women and at 35-39 years in men. The performance in swimming started to deteriorate at 25-29 years in both sexes [17]. Hence, it seems logical that women above 31 and men above 32 years should consider changing to Half Ironman in order to be able to achieve or preserve best results. In summary, this study’s results go along with literature showing that swim performance in recreational Half Ironman athletes seems to be hardest to preserve; that run performance decline is a constant one; and that cycling performance can be preserved up to older age groups.

Other minor changes have been marked red thoughout the whole document.